# Complex tsunamigenic near-trench seafloor deformation during the 2011 Tohoku–Oki earthquake

Kai Zhang [1,2,3], Yanru Wang[1], Yu Luo[1], Dineng Zhao [2,3], Mingwei Wang[2], Fanlin Yang [1,3] ✉ & Ziyin Wu [2] ✉

The near-trench coseismic rupture behaviour of the 2011 Tohoku–Oki earthquake remains poorly understood due to the scarcity of near-field observations. Differential bathymetry offers a unique approach to studying offshore coseismic seafloor deformation but has a limited horizontal resolution. Here we use differential bathymetry estimates with improved horizontal resolutions to investigate near-trench coseismic slip behaviours in the 2011 Tohoku–Oki earthquake. In the main rupture region, a velocity-strengthening behaviour in the shallow fault is observed. By contrast, the seafloor uplift decreases towards the trench, but the trend inverts near the backstop interface outcrop, revealing significant off-fault deformation features. Amongst various competing off-fault effects observed, we suggest that inelastic deformation plays a predominant role in near-trench tsunami excitation. Large trench-bleaching rupture is also observed immediately north of 39°, delimiting the northern extent of the main rupture region. Overall, striking spatial heterogeneity of the shallow rupture behaviour is revealed for the region.

The 2011 Tohoku–Oki earthquake (Mw 9.1) is one of the best instrumentally observed earthquakes in history. Extensive observations reveal shallow rupture behaviours unobserved before, which fundamentally advance our understanding of the megathrust rupture dynamics and the associated tsunamigenesis mechanism. However, for the near-trench region where ruptures can be exceptionally tsunamigenic, critical details of the coseismic seafloor slip remain obscure, and confusion regarding the nature of near-trench fault properties and the associated crustal deformation behaviour persists. Numerous rupture models have been developed based on existing multidisciplinary observations, but their spatial extents and distribution patterns show distinct features in near-trench regions, reflecting fundamentally different fault mechanisms[1,2]. Although a huge large slip approaching the trench has been established for the event[3], no consensus exists on whether the maximum slip occurred directly at the trench axis[4,5] or at some distance down-dip of the trench[6–8]. Whether the large slip peaking at the trench represents a common process in the main rupture region or only reflects site-specific geological conditions is also unclear[9]. In addition, existing models vary considerably on the northern range of the large near-trench slip, which is closely related to the puzzling devastating tsunami along the Sanriku coast[10]. The inversion results based on seismic and geodetic data decline a large slip occurring north of 39°N[11], whereas the inversion incorporating tsunami data generally infer that the large slip extends to a narrow, shallow stripe north of 39°N or even shapes another secondary slip patch in the northern Japan trench in addition to the main rupture region[12–14]. Thus, clarifying the northern extension limit of the main rupture region provides crucial clues for understanding the unexpected immense tsunami runup on the Sanriku coast.

The main reason for the unresolved problems is the scarcity of offshore geodetic observations[15], particularly in near-trench regions. Owing to the challenging nature of direct measurements in abyssal

[1]College of Geodesy and Geomatics, Shandong University of Science and Technology, Qingdao, China. [2]Key Laboratory of Submarine Geosciences, Second Institute of Oceanography, Ministry of Natural Resources, Hangzhou, China. [3]Key Laboratory of Ocean Geomatics, Ministry of Natural Resources, Qingdao, China. ✉e-mail: yang723@163.com; zywu@sio.org.cn

regions, only sparse GPS-acoustic (GPSA) stations recording quantitative coseismic crustal deformation information have been reported[16,17], but all located more than 50 km from the trench. An ocean-bottom pressure (OBP) gauge was deployed ~20 km from the trench[18], but with a large horizontal positioning uncertainty (approximately 20 m).

Complementing the pinpoint GPSA and OBP observations, differential bathymetry offers a unique approach to studying coseismic rupture behaviours. Based on underwater terrain similarity, previous studies[19,20] revealed a huge slip approaching the trench axis by matching the bathymetry before and after the earthquake in the horizontal dimension. Besides, significant seafloor uplift was found on the outermost landward slope. This finding subverts the traditional view of aseismic frontal wedge. Despite these efforts, the coseismic behaviour of the seafloor in near-trench tsunamigenic regions remains poorly resolved primarily because of the low resolution of the horizontal displacement estimates. Given the huge depth involved, seafloor morphological features are distorted by considerable measurement noise and blurred by the low resolution of the data. Thus, previous studies used a large amount of bathymetry data in a single matching to improve the accuracy of the horizontal displacement estimate and account for the interference, leading to a low horizontal estimate resolution of a few tens of kilometres[21–23]. The effectiveness of the treatment is based on an implicit data homogeneity premise and works best when the slip amplitude changes slightly over the area associated with the data used, as in the case reported by Sun et al.[1]. On the contrary, when the local slip is highly heterogeneous, slips of distinct amplitudes associated with different subregions compete with one another in the computation, leading to a compromised estimate of horizontal displacement with degraded accuracy.

In this work, we focus on the links between data size, data homogeneity and estimation accuracy to explore the possibility of improving the resolution of the bathymetry matching result by reducing the size of the bathymetry data in each matching. Reducing the data size shrinks the area of the associated spatial region, which honours data heterogeneity and facilitates an accurate slip characterization when the slip is highly variable. Consequently, seafloor horizontal displacement estimates with improved resolution are obtained whilst retaining the estimation accuracy by considering the effect of athwart bias and outliers in an iterative framework. Improving the horizontal resolution of the bathymetry matching estimates enables us to analyse the coseismic slip behaviour near and across the trench at a finer resolution scale, thereby facilitating the determination of deformation features at spatial resolutions more refined than those in previous works. In addition, a more realistic evaluation of the horizontal estimate uncertainty can be obtained based on the increased estimate sample size on the seaward slope.

## Results

### Study area

Two survey tracks with repeated survey coverage before and after the 2011 Tohoku−Oki earthquake were studied to examine the near-trench coseismic rupture pattern (Fig. 1). The first track locates in the main rupture region and crosses the trench axis at approximately 38.2°N. Kodaira et al.[23]. studied the track by matching the bathymetric data landward of the trench, resulting in a 34 m slip for the landward slope region. On the north and the south sides of the track, the MY101 and MY102 lines had unexpected large slip amplitudes (56 and 69 m, respectively) reaching the trench axis[1,19]. By modelling the bathymetry change for the two tracks with an elastic finite model, Sun et al.[1]. rejected dramatic weakening or strengthening of the shallow fault and indicated that the slip increases uniformly trenchward and peaks at the trench axis. On the northern side of the main rupture region, this work presents track 2, a new processed differential bathymetry profile that crosses the trench axis at 39.05°N. Three tracks on the northern side of

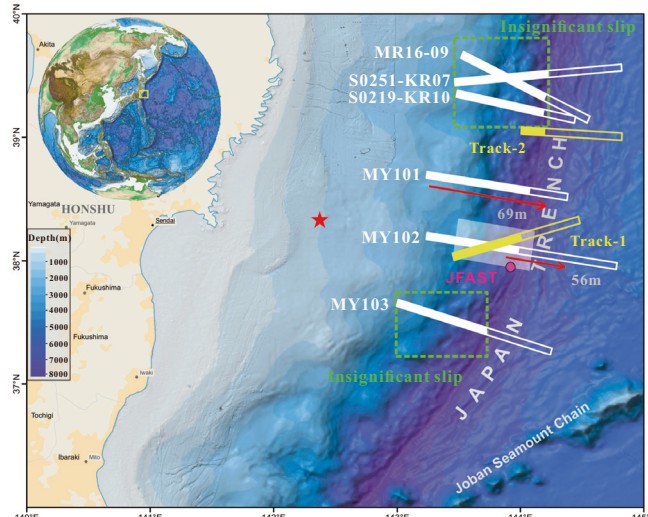

**Fig. 1 | Locations of studied tracks.** The yellow rectangles denote the studied tracks (the boundaries between the solid and hollow portions indicate the locations of the trench axis). The cyan star represents the epicentre. Locations of existing differential bathymetry results are shown in white rectangles[22,23]. The green dashed rectangles indicate the regions where the slip amplitudes were estimated to be insignificant. The purple circle represents the JFAST drilling site where the presence of smectite is proposed to be responsible for the dynamic weakening fault behavior[51–53]. The translucent rectangle represents the central corridor where the seafloor uplift characteristic profile was computed based on tsunami inversion[8].

track 2 were studied by Fujiwara et al.[22,24]., and the slip within the region was statistically insignificant. Their findings opposed the hypothesis that a major near-trench slip extends to offshore Sanriku and contributes to the huge tsunami that hit the Sanriku coast.

### Near-trench coseismic deformation pattern

Track 1 is subdivided along the trench-normal direction with a resolution of 5 km, as shown in Fig. 2a. The horizontal displacement of each segment was estimated using bathymetry correlation matching. In the computation, the window size was set to 5 km to study the along-dip variation of the slip, as shown in Fig. 2d. For comparison, a window size of 10 km was also used for the bathymetry matching, and a variation trend similar to that for 5 km but with a smoothing effect was obtained (Fig. 2e). A complex along-track variation of the seafloor deformation was revealed for the landward side of the trench by combining the horizontal and vertical analysis results. Based on the deformation pattern, the landward slope can be subdivided into two zones with distinctive deformation features, as shown in Fig. 2c.

Zone I is dominated by an evident trenchward decay trend of slip amplitude, which spans ~26 km in the up-dip direction. A huge horizontal displacement at around 50 m was observed in the landward edge of the zone, approximately 40 km from the trench. With a strong along-dip gradient, the displacement amplitude decreases rapidly to a 20 m level at the east end of the zone. Consistent with the horizontal displacement, the vertical seafloor deformation also presents a strong uplift exceeding 10 m at the western end of the zone and decreases generally eastward. The dramatic decrease of slip towards the trench indicates the velocity-strengthening property of the fault friction within the zone. This scenario is fundamentally different from the case of MY 101 and MY102 tracks located nearby, where the slip was reported to increase all the way to the trench axis with a gentle gradient[1].

In zone II, the horizontal displacement stays at a relatively low level and shows a tender seaward tapering trend. By contrast, a distinct deformation pattern of a trenchward increase trend was observed in the vertical dimension. Under the elastic deformation mode, the uplift pattern can be interpreted as a trenchward increase of the slip,

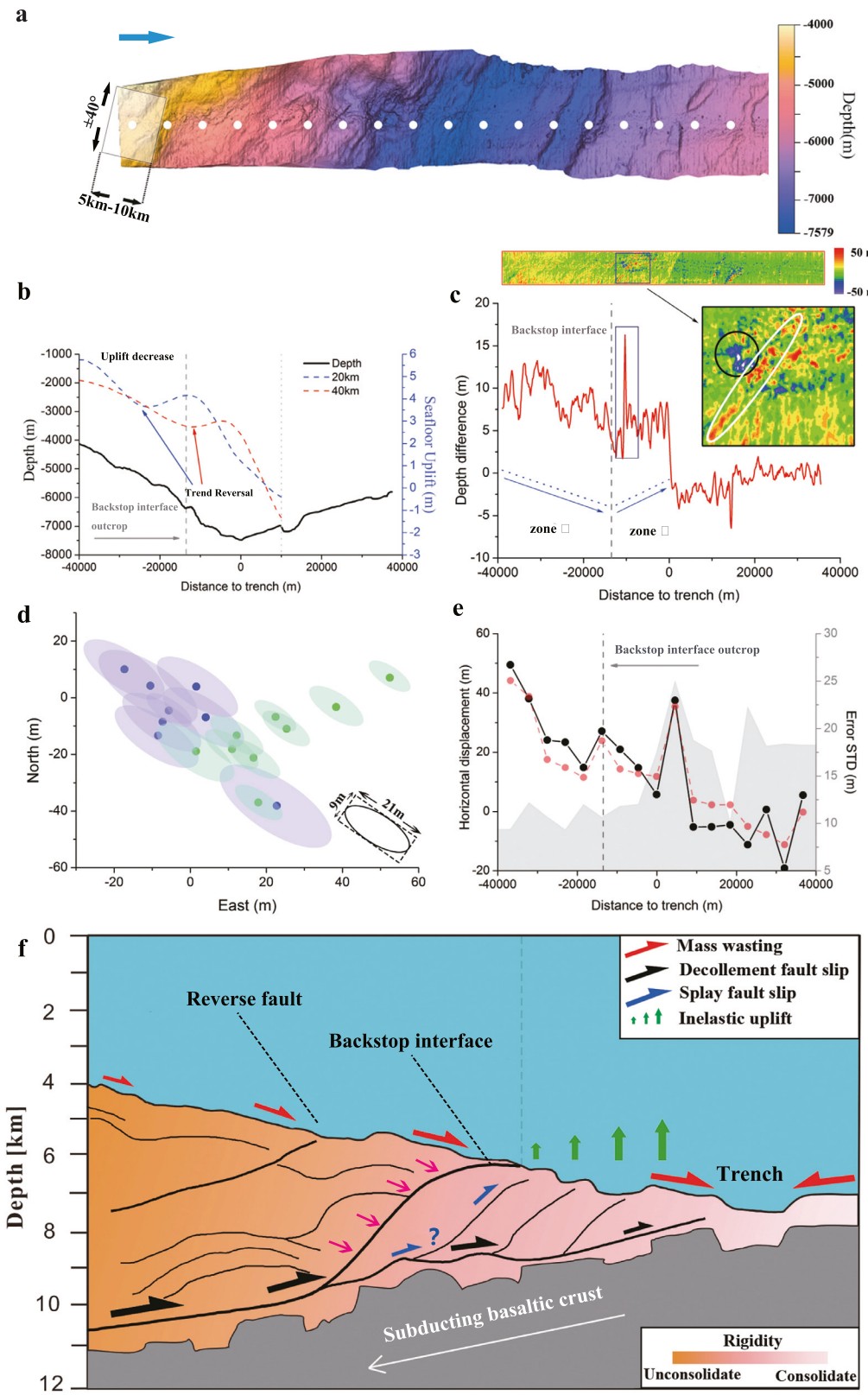

suggesting a coseismic weakening behaviour. However, the trenchward tapering trend of the horizontal displacement disfavours this interpretation. The apparent mismatch of the horizontal and vertical seafloor dislocation suggests the presence of extra physical processes to account for the trenchward increased uplift. The location of the trend inversion corresponds approximately to the outcrop location of the backstop interface, a physical boundary connecting the relatively

harder Cretaceous backstop and the unconsolidated frontal prism. Thus, the inelastic response of the unconsolidated sediment in the frontal prism is a reasonable geological explanation to reconcile the discrepancy. Seaward of the backstop interface, the overriding frontal prism is composed of relatively unconsolidated mass, and its rigidity generally decreases towards the trench. When extruded by the hard backstop, the sediments deform plastically and absorb seismic energy

**Fig. 2 | Differential bathymetry result of track 1. a** Shaded relief map of multi-beam bathymetry data collected in 2012. The white circles denote the centres of the sliding windows. **b** Bathymetry profile along the dip-slip direction. The blue and red dashed curves denote the seafloor uplift profile (corresponding to the translucent rectangle in Fig. 1) computed by tsunami inversion of resolutions of 20 km and 40 km[8]. **c** Bathymetry difference profile computed along the dip-slip direction (positive values for uplift and negative for subsidence). The blue dotted lines denote the best-fit of the depth difference trends landward and seaward of the backstop interface outcrop. Inset shows the local depth variation proximal to the anomalous seafloor uplift peak. The black circle shows the seafloor subsidence likely associated with the submarine landslide. The white ellipse shows the seafloor uplift signature of the trench-parallel ridge structure. **d** Scatter plot of horizontal displacement estimates landward (green) and seaward (purple) of the trench. The variable-size translucent ellipses denote $1\sigma$ uncertainty. **e** Dip-slip component of horizontal displacement estimates with screen window size of 5 km (black curve) and 10 km (red curve). The grey curve shows the uncertainty ($1\sigma$) associated with the black curve. **f** Conceptual model of the coseismic seafloor deformation of track 1. The cross-section structure is motivated by the multichannel seismic reflection data along track MY102[32].

strongly[25,26]. Consequently, a trenchward increase of the seafloor uplift trend coupled with a trenchward slip tapering pattern can be produced under an inelastic model[27,28]. The inelastic deformation of the frontal prism decreases the shallow megathrust slip but converts the rupture energy into the uplift efficiently, which enhances the local tsunami size.

In addition to the inelastic seafloor deformation effect, submarine slope failure is revealed to be an important factor in altering the seafloor uplift pattern, as shown in Fig. 2c. Triggered by the coseismic seafloor motion, unconsolidated sediment on the steep slope is prone to be mobilised by the coseismic seafloor movements[29–31]. Consequently, deposits are left in the deeper region of the lower slope, thereby producing a trenchward increased uplift pattern. A notable sign of slope failure occurrence in zone II is a pronounced peak in seafloor uplift exceeding 15 m. The location of this peak corresponds to a ridge structure oriented parallel to the trench axis. Thus, the striking seafloor uplift is attributed primarily to the slip of the seafloor characterized by the prominent ridge morphology. Nevertheless, the observed uplift requires a horizontal displacement of approximately 50 m because of the megathrust dip angle of around 5° (see Supplementary Fig. 2). By contrast, the corresponding estimated horizontal displacement is only 22 m, which is insufficient to reproduce the large uplift peak, but generally agrees with the uplift level of the surrounding region. A detailed analysis of the bathymetry variation indicates a seafloor subsidence in the steep slope region immediately upslope of the uplift peak. The dipole-like vertical deformation pattern suggests the slope failure contributes to the remarkable seafloor uplift peak.

Moreover, the seismic reflection data[32] indicate that the trench-parallel ridge structure is underlain by a landward dipping ancillary splay fault. The spatial coincidence of the splay fault outcrop and ridge structure suggests they are interconnected. Splay fault activation has been proposed to occur in the 2011 event based on near-field observations[32] and biomarker evidence[33,34]. Here, the geomorphic signature indicates that the activation of the splay fault is also a possible contributor to the anomalous seafloor uplift peak. When the slip is partly redirected from the detachment onto the high-angle splay fault, additional uplift is produced and superimposed on the existing trench-paralleled ridge structure. In this scenario, the activation of the splay fault slip should play an important role in the evolution of the ridge structure.

Further eastwards, the estimated horizontal displacement abruptly increases immediately seaward of the trench axis, indicating a huge coseismic slip approaching the trench axis. However, the uncertainty analysis indicates the corresponding bathymetry matching estimates generally have considerable uncertainty because of the smooth seafloor and perturbation of the significant temporal bathymetry variation. Thus, we interpreted the resulting large slip to be statistically insignificant. For the vertical dimension, the seafloor uplift decreases sharply at the trench axis. Detailed analysis shows that the morphology of the trench axis is remarkably altered by the sediment deposits (Supplementary Fig. 3). Evident seafloor subsidence is present at the seaward slope near the trench. This finding suggests the trench axis shallowness is largely due to the mass transported from the seaward slope. The subsidence feature on the seaward slope is consistent with the tsunami inversion result[8], thereby relevant to the

tsunami excitation. Some notable depth variations are present near the graben slope, which are also likely related to the earthquake-triggered slope failure[35] or artefacts attributed to poor data quality or the bathymetry matching residual. In general, the horizontal displacement and uplift of the seafloor fluctuate around zero on the seaward outer-rise because of the noise perturbation. As the coseismic slip on the seaward trench slope should be negligible, the estimates within this region provide a benchmark to evaluate the uncertainty of the bathymetry matching result. Compared with the previous research, the present study uses a reduced data size in the bathymetry matching to increase the sample size of the estimates on the outer rise, thereby facilitating a more realistic uncertainty estimation. In general, the estimated matching uncertainty on the seaward slope is larger than that on the landward slope (Fig. 2d), primarily because of the relatively smoother seafloor of the seaward slope. Although the horizontal uncertainty of bathymetry matching is inherently large, we conclude that the trenchward decrease of the horizontal displacement amplitude is a robust feature of the landward slope at a minimum.

Located in the northern region, track 2 covers the landward slope for approximately 17 km, which includes the major part of the frontal prism. In this region, the slip amplitudes inverted by tsunami data over different resolutions conflicts with one another because of their large coefficients of variation (Fig. 3b). Moreover, the near-trench slip direction remains controversial; no consensus exists on whether the slip direction is the same as that in the main rupture region offshore Miyagi[4], or rotates anti-clockwise to accommodate the change in the trench direction[36,37]. Thus, unlike the case of track 1, the slip pattern was analysed by projecting the estimated horizontal displacement in the east–west direction. The covered frontal prism moves coherently towards the trench with a maximum horizontal displacement exceeding 20 m (Fig. 3d). Combined with the latitudinal correlation uncertainty (Fig. 3e), the large slip approaching the trench axis is statistically significant. The estimation result provides unambiguous evidence that large coseismic slip extends beyond 39°N. Combined with the findings of Fujiwara et al.[22,24], our differential bathymetry result clearly delineates the northern boundary of the main rupture extension, where the slip propagating from the southern rupture region decreases sharply with a strong gradient in the meridional direction (20 m over 10 km). In the vertical dimension, the apparent seafloor uplift occurs on the landward slope and peaks immediately landward of the trench axis. This pattern is consistent with the horizontal analysis result, indicating the rupture reached the trench axis. Similar to the case in track 1, the seafloor uplift pattern is negatively correlated with the trend of the horizontal displacement, implying the dominant role of off-fault deformation within the frontal prism.

## Discussion

Track 1 has a velocity-strengthening property of the near-trench rupture based on the horizontal displacement estimates resolved at a fine spatial resolution. This property is fundamentally different from that in nearby regions, where the slip increases coherently trenchward and peaks at the trench axis[1,38]. The revealed pattern supports a number of models that place the maximum slip at a distance down-dip the trench; it is also consistent with the sea surface displacement estimated from

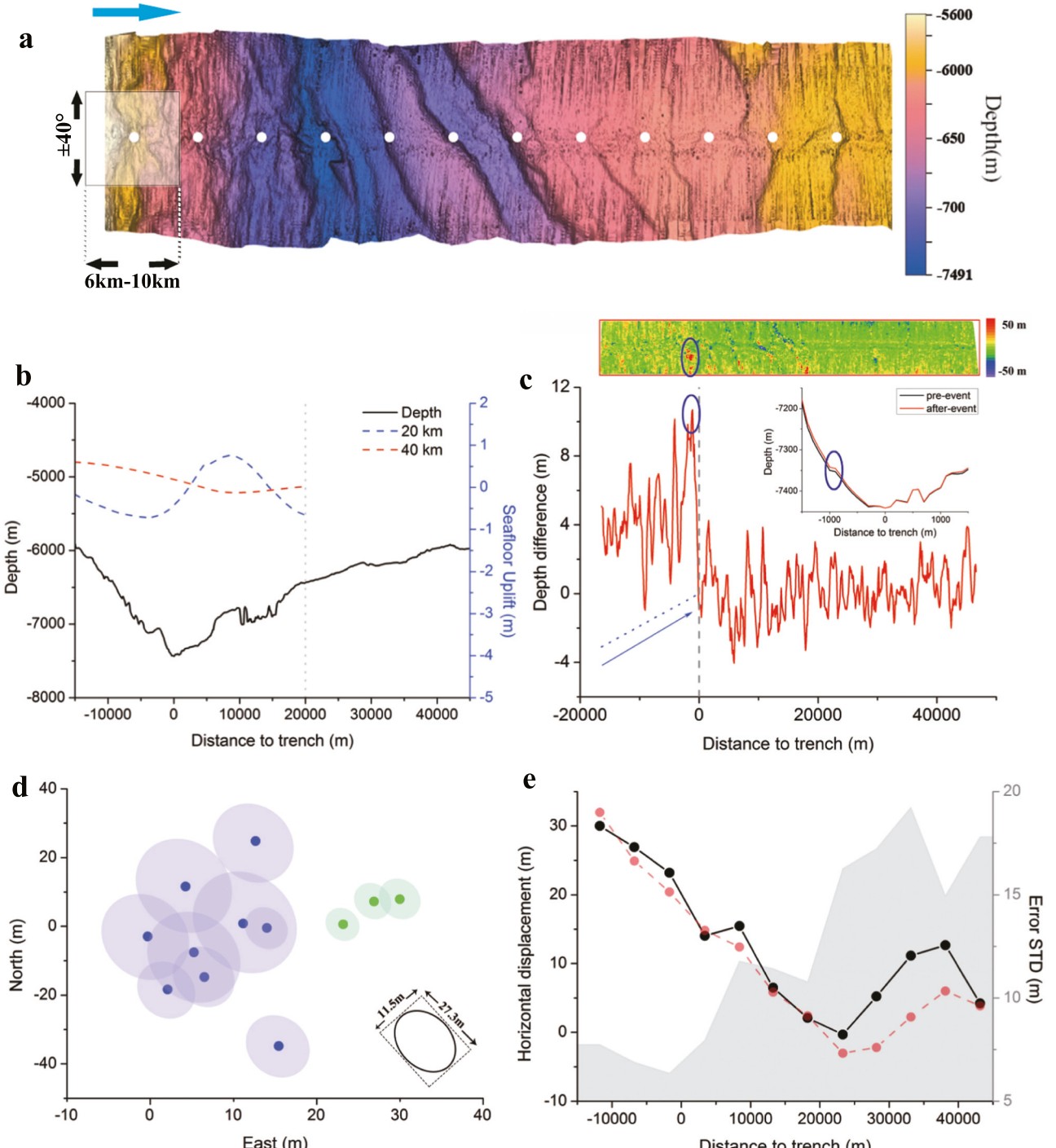

**Fig. 3 | Computation results of track 2. a** Shaded relief map of multibeam bathymetry data collected in 2011. The white circles denote the centres of the sliding windows. **b** Bathymetry profile along the east-west direction. The blue and red dashed curves denote the seafloor uplift profile computed by tsunami inversion of resolutions of 20 km and 40 km[8]. **c** Differential bathymetry acquired by subtracting the 2011 data from the 2010 data. The red curve denotes the bathymetry difference profile computed along the east–west direction. The blue dotted lines denote the best-fit of the depth difference trend landward of the trench. Inset shows the local depth variation with the landslide signature proximal to the trench. **d** Scatter plot of horizontal displacement estimates landward (green) and seaward (purple) of the trench. The variable-size translucent ellipses denote 1σ uncertainty. **e** East-west component of horizontal displacement estimates with screen window sizes of 6 km (black curve) and 10 km (red curve). The grey curve shows the uncertainty (1σ) associated with the black curve. Plotting conventions are in accordance with Fig. 2.

tsunami inversion[8,39]. The low slip level proximal to the trench axis at approximately 38.2°N is broadly consistent with the slip models developed based on multidisciplinary datasets[7] or post-seismic seafloor geodetic inversion results[40].

Another important, intriguing aspect of the near-trench deformation pattern is the inversion of the seafloor uplift trend in the frontal prism with the tapering horizontal displacement, which contradicts the prevailing elastic fault deformation behaviour. Our analysis shows the trend inversion coincides with the backstop interface outcrop. Although the seafloor deformation pattern is also complicated by the mass wasting and potential activation of the splay fault, their effect is constrained in the local spatial region. Thus, the inelastic deformation

of unconsolidated sediments is a plausible interpretation to reconcile the systematic divergence of the horizontal/vertical displacement of the frontal prism. Under the combined influence of the crustal deformation off the plate interface and the velocity-strengthening effect of the megathrust, the seafloor uplift trend decreases towards the trench and then inverses near the backstop interface outcrop. This pattern closely mimics the characteristic profile obtained by tsunami inversion (Fig. 2b)[8], suggesting the revealed seafloor deformation pattern is relevant to the tsunami excitation associated with the central corridor of the main rupture region (Fig. 1).

The similar horizontal/vertical mismatch pattern presented in track 2 further suggests the inelastic effect controlling the frontal prism deformation may be pervasive in the Tohoku–Oki earthquake. Inelastic deformation promotes seafloor uplift, decreases fault slip, and absorbs high-frequency seismic energy, thereby providing an effective mechanism for reconciling the divergence in observations of tsunami earthquakes[26]. Thus, inelastic wedge deformation has been suggested as the cause of the puzzling large run-up along the Sanriku coast during the 2011 Tohoku–Oki earthquake[28]. The results of our study provide direct evidence of the dominant effect of inelastic deformation in the coseismic deformation of the frontal prism, which highlights the substantial oversimplification of the prevailing elastic dominant deformation model in the outer-wedge region. As the most actively deforming section, the coseismic deformation of the outer wedge plays a critical role in controlling tsunami wave height in tsunami earthquakes throughout the world[41]. In a large subduction earthquake when the rupture reaches the trench, the inelastic deformation can play an important role in controlling the seafloor uplift of the frontal prism. Compared with the coseismic slip on the decollement, the inelastic deformation transfers the shallow slip more efficiently to the seafloor uplift, thereby fostering tsunami genesis. Thus, the dominant role of inelastic uplift in the frontal prism deformation revealed in this paper should be relevant to the tsunamigenic potential evaluation in other subduction margins worldwide.

Our findings, when incorporated with previous differential bathymetry studies, reveal the coseismic seafloor deformation pattern is highly variable in the near-trench region. In the central corridor of the main rupture region, the velocity-strengthening and coseismic-weakening fault behaviours[1,42] co-exist within a compact region near the trench, demonstrating a strong along-strike heterogeneity of the rupture. A similar drastic change of shallow frictional behaviours over short distances also occurred in the 2010 Maule earthquake[43,44]. Such heterogeneity plays a critical role in promoting or inhibiting rupture propagation, which is relevant to the generated impulsive tsunami[45]. The strong spatial heterogeneity of the rupture was also shown for the northern Japan Trench. The large trench-breaching slip extends beyond 39°N and decays rapidly further northwards. The delineated northern terminus of the trench-breaching slip agrees with the distribution of the smectite-rich pelagic clay layer on the incoming Pacific Plate[46] and suggests its controlling role in developing a large coseismic slip in the shallowest segment of the megathrust.

Overall, the strong spatial gradient of the slip variation reveals the insufficiency of the existing pinpoint near-field geodetic observations to characterise the highly complicated slip behaviour effectively and highlights the importance of improving the coverage of coseismic seafloor deformation mapping. Differential bathymetry analysis offers a unique way of reconstructing large crustal deformations to fill the gap where GPSA and OBP are absent. The execution of repeated bathymetry surveys along the abundant historical tracks provides a rich data source for backtracing the coseismic slip behaviour. This paper shows that for the near-trench region, the resolution of the horizontal displacement estimate can be improved to a level below 10 km, with a moderate accuracy of approximately 20 m. Thus, the developed algorithm provides an effective way to recover large coseismic slip features in offshore megathrust earthquakes.

## Methods

### Data

The bathymetric data in this study were collected by four different R/Vs (Natsushima, Kaiyo, Yokosuka, Mirai, and Kairei) between 2001 and 2012 (Supplementary Table 1). During such a large time interval, foreshock events and afterslips could also produce observable seafloor displacement, but their effects on the total amplitude of seafloor displacement in the study area were estimated to be 1–2 m at most[11,16]. Thus, the seafloor displacements presented within the differential bathymetry result were assumed mainly coseismic.

After manual editing (deleting the transition of the survey line and removing the invalid data) and outlier filtering (median filter), the bathymetric data were projected to the Gauss plane coordinate and gridded into 20 m intervals (finer than the mean spacing of the raw sounding to retain the short-wavelength morphological features) XYZ format.

### Data correction

Following the paradigm of Kodaira et al.[23], the pre- and post-event datasets were geo-coordinated firstly based on the implicit premise that the bathymetry of the outer-rise region experienced a slight change during the 2011 Tohoku–Oki earthquake. Specifically, the horizontal offset between the two datasets was firstly estimated by correlation matching their overlapped bathymetry on the outer-rise area. After the estimated horizontal offset was removed, the vertical offset was estimated by computing the median of the depth difference values corresponding to the seaward slope region. After the geo-coordination, the coseismic horizontal and vertical displacements on the landward slope were computed by their offsets relative to those on the outer-rise.

The profile was segmented into different blocks with their centre spacing at 5 km along the dip to study the trenchward trend of the slip, considering the compromise between the resolution of the estimates and the accuracy of the bathymetry matching. For track 1, the screen window size for each bathymetry matching was set to 5 km along dip. For track 2, the along-dip width of the screen window was set to 6 km because of the relative sparseness of the bathymetric data cloud. A screen window size of 10 km was also used for verification.

In the computation, evident vertical bias was introduced into the bathymetry data because of the perturbation of sound velocity profile (SVP) error, the absence of tidal correction, and the temporal drift of the ship loading. Thus, the median depth within the screen window was subtracted to remove the absolute depth information, and only the contour information was used for the bathymetry matching. Considering the inherent large noise interference in the edge section of the multibeam swath, bathymetry data with grazing angles larger than 40° were removed in the bathymetry matching.

The directions of the horizontal displacement estimates were poorly constrained owing to the large bathymetry matching uncertainty. The horizontal displacement amplitude was computed for track 1, where the slip direction is well constrained by the existing GPSA observations, by projecting the bathymetry matching results to the slip-dip direction to increase the reliability of the displacement amplitude estimates. As no consensus about the near-trench slip direction has been made yet for the northern Japan trench region, the estimated horizontal displacement of track 2 was projected to the east–west direction to evaluate the slip amplitude variation.

### Horizontal displacement estimation based on bathymetry matching

We subdivide the track with a sliding window along the trench-normal direction to improve the resolution of the coseismic horizontal displacement of the seafloor. Within each window, bathymetry correlation matching between pre- and post-event bathymetry data was independently conducted. Compared with the previous studies, the spatial heterogeneity of the slip and the measurement noise is

honoured by narrowing down the spatial area of the data involved, which contributes to the accurate characterization of the coseismic horizontal displacement. However, reducing the data size used in a single matching generally makes the computation highly sensitive to bias and outliers in the depth data. Thus, we focused on the significant effect of the SVP error and outliers, and developed a new bathymetry matching method based on an iterative framework.

Errors are inevitably introduced in the SVP data because of the spatial–temporal variation of the water properties. Inaccurate SVP information, coupled with considerable water depth, produces crucial transverse artefacts most relevant for perturbing the bathymetry matching estimation. For a given relative error in SVP, the vertical and horizontal relative errors of the ray tracing change monotonously with the grazing angle of the sound signal, resulting in variable ratio coefficients (Supplementary Fig. 1)[47,48]. The SVP-related bias at the edge of the swath is much larger than that in the nadir region; hence, a 'smile' or 'frown' distortion of the seafloor is produced. The vertical error of the ray-tracing computation is generally acceptable for a grazing angle of up to 54°. However, because the multibeam survey tracks are generally in the trench-normal direction, the seafloor slope degree in the cross-track direction is mild, resulting in a small extra vertical error with a secondary effect. Considering these relationships, an empirical algorithm was used to address the effect of the SVP error. Based on an initial estimate, the depth difference values were binned according to their horizontal distances to the track axis (100 m bin size). Then, the median of the depth difference values in each bin was used to represent the local bias due to the SVP error. Then, it was eliminated between the pre- and post-event bathymetric data.

Outliers are pervasive in deep-sea bathymetry data and can seriously interfere with the least square (LS) estimation. An iterative trimming procedure was used to remove the outlier effect on the bathymetry matching computation. Based on an initial estimate, the mean $\mu$ and STD $\sigma$ of the depth difference values [$d_i$, $i = 1,..., n$] were computed as follows using their robust counterparts:

$$\mu = \text{median}(d_i) \tag{1}$$

$$\sigma = 1.4826 \cdot \text{median}(|d_i - \text{median}(d_i)|) \tag{2}$$

A trimming threshold is set to keep $d_i$ only if

$$\mu - c \cdot \sigma < d_i < \mu + c \cdot \sigma, \tag{3}$$

where $c \sim [4, 5]$. On the contrary, the depth difference values exceeding the threshold are regarded as outliers.

A reasonable initial estimate is critical for the iterative correction computation. The least median square (LMS) criterion (4) was used to obtain a robust initial estimate. Compared with the LS criterion (5), estimation based on the LMS criterion generally has lower accuracy but has a high breakdown point (up to 0.5) against outliers[49].

$$(\hat{x}, \hat{y}) = \text{argmin median}\left(\triangle d_i^2\right), \tag{4}$$

$$(\hat{x}, \hat{y}) = \text{argmin} \sum_{i=1}^{n}\left(\triangle d_i^2\right), \tag{5}$$

where $\hat{x}$ and $\hat{y}$ are the estimated horizontal displacements in the east and north directions, respectively, ($\triangle d_l, l = 1, \ldots, s$) is the associated depth differences between the pre- and post-event seafloor, and the argmin operation stands for the argument of the minimum.

Based on the above considerations, the procedure of the iterative bathymetry matching algorithm is summarized as follows:

1. Utilize the LMS matching to calculate the initial estimate ($\hat{x}_0, \hat{y}_0$) based on (4).

2. Calculate the threshold value based on the initial estimate, that is, in Eq. (3), set $c$ to 5, and remove the associated outliers.

3. Conduct LS matching based on (5) to update the estimate, calculate the associated threshold value, that is, in Eq. (3), set $c$ to 4, and remove the outliers.

4. Execute the LS correlation matching with outlier-removed data to update the estimate, and group the bathymetry data according to their horizontal distance to the track.

5. For each group, compute the median of the depth differences, and subtract the athwart bias from the bathymetry data.

6. Execute the LS correlation matching using the bias-removed data to obtain the updated estimate.

7. Iterate steps 3 to 6 until convergence to obtain the final horizontal displacement estimate.

## Evaluation of horizontal displacement estimate uncertainty

Another benefit of increasing the spatial resolution scale of bathymetry matching is the increased number of estimates on the seaward outer rise, which facilitates a more realistic uncertainty evaluation of the obtained horizontal displacement estimates. The fluctuation of the associated estimates provides a clue to evaluate empirically the bathymetry matching uncertainty because the coseismic slip on the seaward trench slope should be negligible.

The accuracy of the bathymetry matching estimate generally increases with the number of depth points used and the roughness of the seafloor, but it decreases with the noise level of the depth measurements[50]. Thus, the ratio between the error STDs of two estimates A and B can be quantified as follows:

$$\frac{\sigma_A}{\sigma_B} = \frac{\sigma_{N,A} \cdot \sigma_{R,B} \cdot s_B}{\sigma_{N,B} \cdot \sigma_{R,A} \cdot s_A}, \tag{6}$$

where $\sigma_{N,A}$ and $\sigma_{N,B}$ are the STD of the noise level, $\sigma_{R,A}$ and $\sigma_{R,B}$ are the seafloor roughness level, and $s_A$ and $s_B$ are the associated bathymetry data size. Given that the noise level is unacknowledged, for a specific estimate ($\hat{x}, \hat{y}$) with pre-event depth measurement ($d_l, l = 1, \ldots, s$), the residual depth difference is used to assess the associated noise level:

$$\sigma_N \propto \sqrt{\frac{\sum_{l=1}^{s}(d_{(\hat{x}, \hat{y}), l} - d_l)^2}{s - 1}}, \tag{7}$$

where ($d_{(\hat{x}, \hat{y}), l}, l = 1, \ldots, s$) is the post-event depths associated with estimate ($\hat{x}, \hat{y}$). Likewise, the roughness of the seafloor is evaluated by computing the statistical fluctuation of the depth within the local search window:

$$\sigma_R \propto \sqrt{\frac{\sum_{l=1}^{s}(\text{STD}(d_{local,l}))^2}{s - 1}}, \tag{8}$$

where $\text{STD}(d_{local,l})$ denotes the sample STD of depths surrounding a particular depth $d_l$ (within a local search radius of 100 m in this paper).

Based on Eq. (8), the weights of each bathymetry matching estimate can be assigned to evaluate its uncertainty level:

$$w_j \propto \frac{\sigma_{R,j}^2 \cdot s_j}{\sigma_{N,j}^2} \tag{9}$$

Based on the LS regression framework, the error variance of the unit weight is computed:

$$\sigma_{X,Unit}^2 = \frac{\sum_{i=1}^{m} w_i \cdot (\hat{x}_{S,i} - \hat{x}_{S,A})^2}{m - 1}, \tag{10}$$

$$\sigma^2_{Y,Unit} = \frac{\sum_{i=1}^{m} w_i \cdot (\hat{y}_{S,i} - \hat{y}_{S,A})^2}{m-1}, \quad (11)$$

where $(\hat{x}_{S,A}, \hat{y}_{S,A})$ is the bathymetry matching estimate computed with the whole seaward slope dataset (the subscript $S$ denotes the seaward slope), and $(\hat{x}_{S,i}, \hat{y}_{S,i}, i = 1,..., m)$ are the fine-scale matching estimates on the seaward slope. With the computed error variance of unit weight, the uncertainty of each horizontal displacement estimate can be evaluated:

$$\sigma_{X,j} = \sigma_{X,Unit} \cdot \sqrt{w_j} \quad (12)$$

$$\sigma_{Y,j} = \sigma_{Y,Unit} \cdot \sqrt{w_j} \quad (13)$$

The bathymetry matching uncertainty in the north–south direction is larger than that in the east–south direction, primarily because of the north–south orientation of the trench and the associated seafloor morphological features. Moreover, the uncertainty in the seaward slope is larger than that in the landward slope due to the relatively smoother seafloor surface. Overall, the uncertainty of the estimates based on reduced data size is compatible with the previous studies (20 m level)[21,22].

### Calculation of vertical displacement

We construct the bathymetry variation profile for the vertical seafloor displacement by binning the depth difference values along the dip-slip direction at a 500 m window scale and computing the median of each bin. The depth difference between pre- and post-event datasets is generally small at the outer-rise region but shows a slight slope phenomenon for track 1. This phenomenon was caused by the depth-dependent bias in the multibeam measurements. Thus, a depth-dependent term was estimated by fitting the bathymetry difference against the depth using the robust fitting procedure to account for the biased trend in the depth difference profile (Supplementary Fig. 4). Under the Gaussian distribution assumption, the uncertainty associated with a specific value in the depth difference profile was evaluated:

$$\sigma = 1.4826 \cdot \mathrm{median}(|d_i - \mathrm{median}(d_i)|)/\sqrt{s}, \quad (14)$$

where $(d_i, i = 1, ..., s)$ is the depth difference values within the screen window, and $s$ is the associated value number. Generally, the uncertainty ($1\sigma$) associated with the depth difference profile is within 2 m (Supplementary Fig. 5).

## Data availability

The bathymetry data used in this study are available from the JAMSTEC database site (http://www.godac.jamstec.go.jp/darwin/e). Source data are also provided with this paper. Source data are provided with this paper.

## Code availability

The Matlab® codes used for correlation matching can be accessed at the online repository (https://codeocean.com/capsule/5251089/tree).

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

## Acknowledgements

We are grateful to the Japan Agency for Marine-Earth Science and Technology (JAMSTEC) for providing the bathymetry data used in this study.

This work was supported in part by the National Key R&D Program of China (grant 2022YFC2806600 & 2022YFC2806602) to Z.W., K.Z., and M.W., in part by the National Natural Science Foundation of China to Z.W. (grant 41830540) and F.Y. (grant 41930535), in part by the Special Projects for Promoting High-Quality Economic Development (Marine Economic Development) in Guangdong Province (grant GDNRC[2023] 42) to F.Y., in part by Natural Science Foundation of Shandong Province (grant ZR2020MD084) to K.Z., in part by Shandong University of Science and Technology Research Fund (grant 2019TDJH103) to F.Y., and in part by Key Laboratory of Ocean Geomatics, Ministry of Natural Resources of China (grant 2021B05) to D.Z.

## Author contributions

K.Z. and Z.W. contributed to developing the main idea. Y.W., D.Z., and M.W. processed the data. K.Z., Y.W., and Y.L. carried out the analysis. F.Y. and Z.W. oversaw the study. K.Z. and Y.L. edited the manuscript. All authors commented on the paper.

## Competing interests

The authors declare no competing interests.
