## [Peer Review File · Nature Communications]

Complex tsunamigenic near-trench seafloor deformation during the 2011 Tohoku–Oki earthquakeEditorial Note: Parts of this Peer Review File have been redacted as indicated to remove third-party material where no permission to publish could be obtained.

REVIEWER COMMENTS

Reviewer #1 (Remarks to the Author):

Review for NCOMMS-23-04328

Recognising the complex near-trench coseismic deformation in 2011 Tohoku-Oki earthquake: insight from differential bathymetry analysis with improved resolution
Written by Kai Zhang, Fanlin Yang, Ziyin Wu

In this paper, the authors examined the seafloor displacement caused by the 2011 Tohoku-Oki earthquake. The resolution of the bathymetric data has been dramatically improved. However, the authors have carefully processed the data to avoid errors and variations were examined in detail. The resolution seems to be ultimately affected by the quality of the bathymetric data obtained before the earthquake and does not appear to have been

biased that could be introduced in the small interval analysis, and the data are of the expected standard.

The results show that the horizontal displacement gradually decreases toward the trench axis. Therefore, the authors suggest that the rupture of the shallow plate boundary fault near the trench axis is considered to be caused by a velocity strength-type mechanism. On the other hand, the vertical displacement increases toward the trench. The authors suggest that these phenomena cannot be explained by elastic upper plate deformation, but by inelastic frontal wedge deformation.

It is interesting to note that the results for horizontal displacement near the trench axis are different from those of the previous study by Sun et al. (2017). Taken together with the results of this previous study, this suggests that there is a strong regional variability in the behavior of fault rupture and crustal deformation near the trench. This is an interesting paper for the discussion of the behavior of fault rupture and crustal deformation near trenches.

Regarding references, the possibility of inelastic crustal deformation was also discussed by Fujiwara et al. (2017), and Kodaira et al. (2020). Attempts to study seafloor displacement in small sections have been done previously by Maksymowicz et al. (2017, Scientific Reports) and Fujiwara (2021, Frontiers in Earth Science), so the reviewer thinks the authors should refer to them.

The development of analysis methods of differential bathymetry is important as a complementary technique for understanding the co-seismic and post-seismic seafloor deformation over the rupture area. The author's proposed analytical method described in the Supplementary Material is significant because it is thought to be an important finding that will contribute to improving the accuracy of observations and is thought to be of interest to specialists.

As for the detailed description of the methods for the work to be reproduced, the reviewer could not understand the following points. The reviewer would appreciate it if the authors could consider these points and improve the manuscript.

Lines 442-449, the authors state that the seaward slope region is used as the reference for correction and the landward variation is examined. Since the analysis is done at several locations in the seaward slope region, where is it used for the reference? Or, it is not clear how the authors conclude the seafloor displacement on the seaward slope.

Regarding the estimation and removal of bias due to SVP errors described in Lines 479-493, both measurements of pre-earthquake and post-earthquake surveys should contain SVP errors, so the depth difference is a composite of SVP errors. Can SVP errors really be removed by this method? At any rate, it seems that the distortion of differential bathymetry can be removed, but if the estimated SVP error-related bias is subtracted from the depth difference, isn't the vertical displacement not obtained?

Minor comments

Figure 1: Purple circle is missing.

The MR16-09 track in Figure 1 is not listed in References 19-22. Perhaps it is taken from Fujiwara (2021, *Frontiers in Earth Science*)?

Figure 2(d) and 3(d): It is not clear which circle symbols indicate the location of the results. The distribution is supposed to be meaningful, so please make it clear.

Line 520: Step 5 is missing. Perhaps misnumbered?

Figure S2: (a) and (b) are not labeled in the figure.

Figure S5 is not cited in the paper.

Reviewer #2 (Remarks to the Author):

This paper, titled "Recognising the complex near-trench coseismic deformation in 2011 Tohoku-Oki earthquake: insight from differential bathymetry with improved resolution" describes measurements of seafloor deformation by the 2011 Tohoku-oki earthquake using the differential bathymetry before and after the earthquake. Comparison of bathymetry data collected before and after the event provides useful information for crustal movement. However horizontal resolution of seafloor movement is generally low because depth data for relatively large area are used for the comparison. They used bathymetry data from narrow regions to increase resolution and compared data before and after the event. Detailed difference of seafloor topography was obtained by their analysis using data captured by multi-narrow beam echo sounders equipped to research vessels. They found there are two patterns for seafloor deformation in the study area, and interpreted that one pattern corresponds to velocity strengthening of the fault friction and another pattern shows inelastic deformation during the event. There is a possibility that the inelastic uplift in the frontal prism deformation is related to generation of large tsunami off Sanriku region. In the Supplementary Materials, data and a method for calculation of seafloor deformation in the study are described in detail.

This manuscript newly proposes an useful and unique method to estimate seafloor deformation by large interplate earthquake using data from multi narrow beam echo sounder and gives new evidence for consideration in deformation near a trench at occurrence of large events. Description of their data processing including elimination of outliers from the dataset and estimation of errors in Supplementary Materials are useful. The manuscript is concluded to be worthy for the publication. However I believe that minor revision is needed for publication to understand their arguments clearly.

Major comments

It is argued that an increase of the seafloor uplift to the trench with a slip tapering pattern was generated under an inelastic deformation. This interpretation is thought to be important in consideration of tsunami generation. For better understanding the argument, detailed explanation of deformation occurred by the Tohoku-oki earthquake in the frontal zone near the trench in relation to tsunami generation is useful.

For comparison of bathymetry data before and after the event, dividing smaller region is thought to give spatially higher resolution. In this paper, the regions have a spatial scale of approximately 5 km. Please explain what determines a size of regions which are used for comparison.

It is reported that the slip distribution from Track-1 is different from those from MY-101 and MY 102 tracks. However MY 102 seems to be overlapped Track-1 in Fig. 1. Detailed explanation is needed for interpretation.

Seafloor deformation during a large interplate earthquake strongly corresponds to generation of the tsunami. For the 2011 Tohoku-oki earthquake, it is known that large slip near the trench contributed to generation of the large tsunami. In the discussion sections, it is useful to add systematic discussion for a relation between the results of this study and tsunami generation at occurrence of the event.

Bathymetry data of incident angles greater than 40 degrees were removed for data processing. Please explain quantitatively the reason of removals of the data.

Elimination of outlier from the dataset seems to be a reasonable and usual method for data with gaussian distribution. For clear understanding, please explain "arg min" in equations (4) and (5). I guess that "arg min" means that picking out a single point which gives minimum value. Short explanation of mathematical term is helpful for readers. In addition, it is difficult to understand how estimate errors of difference of topography before and after the event. please explain a concept how to estimate errors.

There are some figures not to refer in the text. All figures should be referred in the text. It is better that all figures are mentioned in appropriate part of the text.

Although I may not make a good judge, I feel that improvement of English should be needed. Some sentences are not clear for me.

Minor comments

Near-trench coseismic deformation pattern

p.9 L 14: Zone I and II should be clearly defined.

p.9 L 22: Is citation of Fig. 3(b) appropriate ?

Figure 1

Purple circle which indicate JFAST site is difficult to see. There is no description for region surrounding by green dash curves.

Figure 3 (b)

There seems to be no explanation for curves annotated by 20 km and 40 km.

Revision list according to the comments

Title: Complex near-trench coseismic deformation in the 2011 Tohoku-Oki earthquake: insight from improved-resolution differential bathymetry (NCOMMS-23-04328)

Firstly, we would like to sincerely thank the reviewers for the invaluable comments that have certainly improved the quality of the manuscript significantly. In the revised manuscript, a number of inappropriate expressions were corrected following the suggestions of the reviewers. Besides, four new references were supplemented (Maksymowicz et al., 2017; Fujiwara, 2021; Wu et al., 2021; Jara-Muñoz et al., 2022). The following is a brief revision list according to the comments from reviewers. The corresponding changes are also marked in red in the revised manuscript.

Reviewer #1:

Generall comments

In this paper, the authors examined the seafloor displacement caused by the 2011 Tohoku-Oki earthquake. Compared to previous studies, smaller sub-sections were analyzed and spatial variations were examined in detail. The resolution seems to be ultimately affected by the quality of the bathymetric data obtained before the earthquake and does not appear to have been dramatically improved. However, the authors have carefully processed the data to avoid errors and biases that could be introduced in the small interval analysis, and the data are of the expected standard.

The results show that the horizontal displacement gradually decreases toward the trench axis. Therefore, the authors suggest that the rupture of the shallow plate boundary fault near the trench axis is considered to be caused by a velocity strength-type mechanism. On the other hand, the vertical displacement increases

toward the trench. The authors suggest that these phenomena cannot be explained by elastic upper plate deformation, but by inelastic frontal wedge deformation.

It is interesting to note that the results for horizontal displacement near the trench axis are different from those of the previous study by Sun et al. (2017). Taken together with the results of this previous study, this suggests that there is a strong regional variability in the behavior of fault rupture and crustal deformation near the trench. This is an interesting paper for the discussion of the behavior of fault rupture and crustal deformation near trenches.

The development of analysis methods of differential bathymetry is important as a complementary technique for understanding the co-seismic and post-seismic seafloor deformation over the rupture area. The author's proposed analytical method described in the Supplementary Material is significant because it is thought to be an important finding that will contribute to improving the accuracy of observations and is thought to be of interest to specialists.

Major comments

1. Regarding references, the possibility of inelastic crustal deformation was also discussed by Fujiwara et al. (2017), and Kodaira et al. (2020). Attempts to study seafloor displacement in small sections have been done previously by Maksymowicz et al. (2017, Scientific Reports) and Fujiwara (2021, Frontiers in Earth Science), so the reviewer thinks the authors should refer to them.

Reply: Thanks for the advice. In the first draft of the manuscript, (Fujiwara et al., 2017), and (Kodaira et al., 2020) were referred to. We have supplemented the following references in the revised manuscript: (Maksymowicz et al., 2017; Fujiwara, 2021).

2. As for the detailed description of the methods for the work to be reproduced, the reviewer could not understand the following points. The reviewer would appreciate it if the authors could consider these points and improve the manuscript.

Lines 442-449, the authors state that the seaward slope region is used as the reference for correction and the landward variation is examined. Since the analysis is done at several locations in the seaward slope region, where is it used for the reference? Or, it is not clear how the authors conclude the seafloor displacement on the seaward slope.

Reply: Because the coseismic deformation of the seaward slope is supposed to be negligible, the bathymetry data on the seaward slope region is treated as a whole in the bathymetry matching. Afterward, the correlation result is considered to be systematic bias for the entire track, and is subtracted from the improved-resolution matching results. Thus, the reference computation procedure is similar to that of (Fujiwara et al., 2011; Fujiwara, 2021).

3. Regarding the estimation and removal of bias due to SVP errors described in Lines 479-493, both measurements of pre-earthquake and post-earthquake surveys should contain SVP errors, so the depth difference is a composite of SVP errors. Can SVP errors really be removed by this method? At any rate, it seems that the distortion of differential bathymetry can be removed, but if the estimated SVP error-related bias is subtracted from the depth difference, isn't the vertical displacement not obtained?

Reply: The offsets in horizontal and vertical dimensions were computed separately. In bathymetry matching, the SVP-related bias curve is computed to capture the transverse artefact pattern (the 'smile' or 'frown' distortion effect). Whilst in the along-track direction, the bias curve is fixed for all pings of bathymetry data within the sliding window. Thus, the transverse distortion can be largely alleviated, but the contour information is retained for the bathymetry matching. By contrast, in the vertical displacement computation, the seafloor uplift magnitudes of the landward slope are computed by their offsets relative to those on the outer-rise. Thus, the disturbance of SVP-related error has been eliminated to a large extent (if the depth remains unchanged along the track, the SVP-related error should be completely eliminated). In the computation result, only the bias due to the interplay between the along-track depth variation and the SVP error remains. To this end, a depth-dependent

term is introduced to empirically characterize the bias in the multibeam measurements that couples with the depth variation. The calibration effect is shown in Fig. S4 (Supplementary Fig. 4).

Minor comments

1. Figure 1: Purple circle is missing.

Reply: Thanks for pointing out this mistake. The purple circle has been added in the figure.

2. The MR16-09 track in Figure 1 is not listed in References 19-22. Perhaps it is taken from Fujiwara (2021, *Frontiers in Earth Science*)?

Reply: Yes, it is taken from (Fujiwara, 2021). We have supplemented the reference in the revised manuscript.

3. Figure 2(d) and 3(d): It is not clear which circle symbols indicate the location of the results. The distribution is supposed to be meaningful, so please make it clear.

Reply: We are sorry for this ambiguity. The spatial location of the matching result is indicated by the white circles in Fig. 2(a) and Fig. 3(a). In Fig. 2(d) and Fig. 3(d), the solid circles denote the matching results (estimated horizontal displacements), and the variable-size translucent ellipses denote the corresponding 1σ uncertainty. The description was rephrased in the revised manuscript.

4. Line 520: Step 5 is missing. Perhaps misnumbered?

Reply: Thanks for pointing out this typo. It was corrected in the revised manuscript.

5. Figure S2: (a) and (b) are not labeled in the figure.

Reply: The panel labels have been supplemented in Figure S2.

6. Figure S5 is not cited in the paper.

Reply: Thanks for pointing out this mistake. The citation has been supplemented.

Reviewer #2:

General comments

This paper, titled "Recognising the complex near-trench coseismic deformation in 2011 Tohoku-Oki earthquake: insight from differential bathymetry with improved resolution" describes measurements of seafloor deformation by the 2011 Tohoku-oki earthquake using the differential bathymetry before and after the earthquake. Comparison of bathymetry data collected before and after the event provides useful information for crustal movement. However horizontal resolution of seafloor movement is generally low because depth data for relatively large area are used for the comparison. They used bathymetry data from narrow regions to increase resolution and compared data before and after the event. Detailed difference of seafloor topography was obtained by their analysis using data captured by multi-narrow beam echo sounders equipped to research vessels. They found there are two patterns for seafloor deformation in the study area, and interpreted that one pattern corresponds to velocity strengthening of the fault friction and another pattern shows inelastic deformation during the event. There is a possibility that the inelastic uplift in the frontal prism deformation is related to generation of large tsunami off Sanriku region. In the Supplementary Materials, data and a method for calculation of seafloor deformation in the study are described in detail.

This manuscript newly proposes an useful and unique method to estimate seafloor deformation by large interplate earthquake using data from multi narrow beam echo sounder and gives new evidence for consideration in deformation near a trench at occurrence of large events. Description of their data processing including elimination of outliers from the dataset and estimation of errors in Supplementary Materials are useful. The manuscript is concluded to be worthy for the publication. However I believe that minor revision is needed for publication to understand their

arguments clearly.

Major comments

1. It is argued that an increase of the seafloor uplift to the trench with a slip tapering pattern was generated under an inelastic deformation. This interpretation is thought to be important in consideration of tsunami generation. For better understanding the argument, detailed explanation of deformation occurred by the Tohoku-oki earthquake in the frontal zone near the trench in relation to tsunami generation is useful.

Reply: Thanks for the advice. We have supplemented the following description of the relationship between seafloor deformation and tsunamigenesis in the Result Section:

The inelastic deformation of the frontal prism decreases the shallow megathrust slip but converts the rupture energy into the uplift efficiently, which enhances the local tsunami size.

We hope this expression is appropriate.

2. For comparison of bathymetry data before and after the event, dividing smaller region is thought to give spatially higher resolution. In this paper, the regions have a spatial scale of approximately 5 km. Please explain what determines a size of regions which are used for comparison.

Reply: In each bathymetry matching computation, a sliding window was used to screen the bathymetry data, thereby controlling the spatial scale of the bathymetry matching result. We iteratively reduce the size of the sliding window to explore its lower limit. In the process, the consistency of the matching results on the seaward slope is used to evaluate the validity of the region size parameterization. For track 1, the lower limit of the along-dip width of the screen window is evaluated to be 5 km. While for track 2, the along-dip window width was set to 6 km because of the relative sparseness of the bathymetric data cloud. For verification, a screen window size of 10 km was also used for comparison. By comparing the estimates of different resolutions, we interpreted that the trenchward decrease of the horizontal displacement amplitude

is a robust feature of the landward slope.

3. It is reported that the slip distribution from Track-1 is different from those from MY-101 and MY 102 tracks. However MY 102 seems to be overlapped Track-1 in Fig. 1. Detailed explanation is needed for interpretation.

Reply: Yes, the two tracks are overlapped with each other. Previous studies used large sizes of bathymetry data in a single matching process. Specifically, data of the track corresponding to the landward slope was used as a whole in the correlation matching. Thus, the slip magnitudes of MY-101 and Track-1 are 56 m and 34 m, respectively (Kodaira et al., 2020). Each estimation slip magnitude is considered a compromised estimate of slip associated with different subregions. The difference in estimation results suggests the evident spatial heterogeneity of the fault slip properties. In our study, the data of Track-1 is segmented into different regions with smaller spatial scales. For comparison, we use the overlapped region between two tracks for the correlation matching analysis and obtain a slip magnitude of 21 m. The result is generally consistent with the previous matching result of Track-1 (34 m), but is much smaller than the previous result of MY-101 (56 m). We ascribe this difference to the strong spatial gradient of the slip magnitude and the relatively large estimation uncertainty (~20 m). Generally, both the results in Kodaira et al. and our study consistently suggest the strong spatial heterogeneity of the slip magnitude within the main rupture region near the trench.

4. Seafloor deformation during a large interplate earthquake strongly corresponds to generation of the tsunami. For the 2011 Tohoku-oki earthquake, it is known that large slip near the trench contributed to generation of the large tsunami. In the discussion sections, it is useful to add systematic discussion for a relation between the results of this study and tsunami generation at occurrence of the event.

Reply: Thanks for the valuable comments. Under the traditional elastic deformation model, the tsunami generation efficiency of the shallow fault slip is often limited by

the typically gentle slope of the seafloor. In contrast, the inelastic deformation model provides a more efficient mechanism for tsunami generation (Ma & Nie, 2019). Thus, we have supplemented the following discussion about the relationship between seafloor deformation and tsunami genesis:

In a large subduction earthquake when the rupture reaches the trench, the inelastic deformation can play an important role in controlling the seafloor uplift of the frontal prism. Compared with the coseismic slip on the decollement, the inelastic deformation transfers the shallow slip more efficiently to the seafloor uplift, thereby fostering tsunami genesis.

We hope this expression is appropriate.

5. Bathymetry data of incident angles greater than 40 degrees were removed for data processing. Please explain quantitatively the reason of removals of the data.

Reply: The basic consideration of bathymetry data trimming is the compromise between the efficiency of data utilization and measurement accuracy. In previous studies (Fujiwara et al., 2011; 2017, e.g.), bathymetry data of a long survey line landward of the trench was used simultaneously in a single matching process, and only the data within beam angle of $45^\circ (\pm 22.5^\circ)$ among 120° swath were used for analysis. By contrast, we reduced the length of the survey line involved in a single matching to improve the resulting resolution. Thus, enlarging the data utilization scope in the across-track direction is helpful. Due to the significant interference of SVP error in deep-water settings, the edge section of the multibeam bathymetry data is with larger error. Quantitatively, the error in depth increases nonlinearly with the incidence angle of the acoustical signal (Fig S1a). Generally, bathymetry data within $\pm 50^\circ$ is often thought to be reasonably acceptable in mapping applications, given the effect pattern of the SVP error. In reciprocating survey mode, the bathymetry data near $\pm 45^\circ$ can be used as a reference to correct the SVP error (Kammerer, 2000; Wu et al., 2021), as shown in Fig. 1 (below in this file). From a conservative viewpoint, bathymetry data of incident angles greater than 40 degrees were removed for data processing in this

study.

Fig. 1 The SVP effect correction based on partly overlapped bathymetry data, the blue and green colors denote the data with opposite track directions. (a) the raw data (seven swaths); (b) bathymetry after sound refraction artifacts are corrected.

6. Elimination of outlier from the dataset seems to be a reasonable and usual method for data with gaussian distribution. For clear understanding, please explain "arg min" in equations (4) and (5). I guess that "arg min" means that picking out a single point which gives minimum value. Short explanation of mathematical term is helpful for readers. In addition, it is difficult to understand how estimate errors of difference of topography before and after the event. please explain a concept how to estimate errors.

Reply: Thanks for pointing out this inappropriate expression. The operation "arg min" stands for the argument of the minimum, the corresponding description has been supplemented.

In the bathymetry matching, for a particular patch of seafloor P in the pre-event dataset, the most similar patch Q with the same size is searched in the post-event

dataset. Afterward, the difference of topography is computed as:

$$d_i = z_{p,i} - z_{q,i} \quad (1)$$

where, $\{z_{p,i}, i = 1 \sim n\}$ and $\{z_{q,i}, i = 1 \sim n\}$ represent the depth values of regions P and Q, respectively. Thus, to resist the disturbance of the potential outliers, the mean μ and STD σ of the depth difference values are computed using their robust counterparts:

$$\mu = \text{median}(d_i) \quad (2)$$

$$\sigma = 1.4826 \cdot \text{median}(|d_i - \text{median}(d_i)|) \quad (3)$$

Under the Gaussian distribution approximation, a trimming threshold is set to keep d_i only if

$$\mu - c \cdot \sigma < d_i < \mu + c \cdot \sigma \quad (4)$$

where $c \sim [4,5]$ (Olive, 2008).

Considering the accuracy of the initial horizontal offset estimate, a relatively loose threshold assignment ($c = 5$) is used in step 2. While in step 3, the parameter c is set to be 4. Thus, based on the recursive adaption, the impact of outliers is supposed to be accounted for.

7. There are some figures not to refer in the text. All figures should be referred in the text. It is better that all figures are mentioned in appropriate part of the text.

Reply: Thanks for pointing out the mistake. We have supplemented the citation of Fig. S5 in the revised manuscript.

8. Although I may not make a good judge, I feel that improvement of English should be needed. Some sentences are not clear for me.

Reply: Thanks for the advice. We have tried to improve the language by seeking help from a professional proof reading service (Fig. 2 in this file).

[REDACTED]

Fig. 2 The language review certification of the revised manuscript.

Minor comments

1. Near-trench coseismic deformation pattern

Page 9 line 14: Zone I and II should be clearly defined.

Reply: Zone I and II are defined in Fig.2 (c)., the figure citation has been added in the revised manuscript (Page 6, line 16).

2. Page 9 line 22: Is citation of Fig. 3 (b) appropriate ?

Reply: On line 22 of page 9, we intend to show the slip amplitude inverted by tsunami data over different resolutions, which are delineated in Fig. 3(b) (the red and blue dashed curves). After supplementing the explanation in the legend of Fig. 3(b), the citation seems to be appropriate.

3. In Figure 1, purple circle which indicate JFAST site is difficult to see. There is no

description for region surrounding by green dash curves.

Reply: Thanks for pointing out this inappropriate expression. In the revised manuscript, the purple circle is highlighted, and the description for the green dash curves has been supplemented.

4. Figure 3 (b), there seems to be no explanation for curves annotated by 20 km and 40 km.

Reply: The description for these two curves has been supplemented in the legend of Fig. 3.

Reference

Fujiwara, T. et al., The 2011 Tohoku-Oki earthquake: Displacement reaching the trench axis. *Science* 334, 1240–1240 (2011).

Fujiwara, T. et al. Seafloor displacement after the 2011 Seafloor Displacement after the 2011 Tohoku-Oki earthquake in the Northern Japan trench examined by Repeated Bathymetric surveys. *Geophys. Res. Lett.* 44(23), 11833–11839 (2017).

Fujiwara T. Seafloor Geodesy From Repeated Multibeam Bathymetric Surveys: Application to Seafloor Displacement Caused by the 2011 Tohoku-Oki Earthquake. *Front. Earth Sci.* 9:667666 (2021).

Jara-Muñoz, J., Melnick, D., Li, S. et al. The cryptic seismic potential of the Pichilemu blind fault in Chile revealed by off-fault geomorphology. *Nat Commun* 13, 3371 (2022).

Kammerer E. A new method for the removal of refraction artifacts in multibeam echosounder systems.[D]. The University of New Brunswick (2000).

Kodaira, S., Fujiwara, T., Fujie, G., Nakamura, Y., & Kanamatsu, T. Large Coseismic Slip to the Trench During the 2011 Tohoku-Oki Earthquake. *Annu. Rev. Earth Planet. Sci.* 48, 321–343 (2020).

Ma, S., & Nie, S. "Dynamic Wedge Failure and Along-Arc Variations of Tsunamigenesis in the Japan Trench Margin." *Geophys. Res. Lett.* 46 (2019).

Maksymowicz, A., Chadwell, C. D., Ruiz, J., Tréhu, A. M., Contreras-Reyes, E., Weinrebe, W., et al. Coseismic Seafloor Deformation in the Trench Region during the Mw8.8 Maule Megathrust Earthquake. *Sci. Rep.* 7, 45918 (2017).

Olive, D. J., 2008. Applied robust statistics. <http://www.math.siu.edu>.

Wu, Z., et al. *High-resolution Seafloor Survey and Applications.*, (Springer, Singapore, 2021).

REVIEWERS' COMMENTS

Reviewer #1 (Remarks to the Author):

In response to the points raised by the reviewers, the authors have filled in some missing parts of the description.

Therefore, the reviewer thinks the manuscript has been improved to a certain level.

Minor points

Page 26, Line 14: "Gou, F." -> "Fujie, G."

Reviewer #2 (Remarks to the Author):

This revised paper, titled "Complex near-trench coseismic deformation in the 2011 Tohoku-Oki earthquake: insight from differential bathymetry" describes measurements of seafloor deformation by the 2011 Tohoku-oki earthquake using the differential bathymetry before and after the earthquake. They compared bathymetry data collected before and after the earthquake and obtain interesting information for crustal movement. Bathymetry data captured by multi-narrow beam echo sounders equipped to research vessels were divided in narrow regions to increase resolution, and detailed difference of seafloor topography was obtained. There is a possibility that the inelastic uplift in the frontal prism deformation is related to generation of large tsunami off Sanriku region. In the Supplementary Materials, data and a method for calculation of seafloor deformation in the study are described in detail.

The requirements for the previous version have been fulfilled. They added explanation of deformation by the 2011 Tohoku-oki earthquake in the Result Section. Criterion for dividing the regions for estimation of deformation by the event and difference of slip amounts between the profiles were reasonably explained. Description about the relationship between seafloor deformation and tsunami genesis was additionally described in the Discussion Section. The data processing about removal of data with large incident angles and mathematical description were also well explained. Other comments for the previous manuscript have been reflected.

The revised manuscript has much improvement and is thought to be understandable for readers. The useful and unique method is newly proposed to estimate seafloor deformation by a large interplate earthquake. In addition, the authors give new evidence for consideration in deformation near a trench during large events. Their data processing is well described in Supplementary Materials. I feel the manuscript is acceptable for the publication.

Revision list according to the comments

Title: Complex near-trench coseismic deformation in the 2011 Tohoku-Oki earthquake: insight from improved-resolution differential bathymetry (NCOMMS-23-04328B)

Firstly, we would like to sincerely thank the reviewers for the invaluable comments that have certainly improved the quality of the manuscript significantly. In the revised manuscript, a mistake in reference was corrected. The following is a brief revision list according to the comments from reviewers.

Reviewer #1:

Generall comments

In response to the points raised by the reviewers, the authors have filled in some missing parts of the description.

Therefore, the reviewer thinks the manuscript has been improved to a certain level.

Reply: Thanks for the constructive comments.

Minor comments

1. Page 26, Line 14: "Gou, F." -> "Fujie, G."

Reply: Thanks for pointing out this mistake. It was corrected in the revised manuscript.

Reviewer #2:

Generall comments

This revised paper, titled "Complex near-trench coseismic deformation in the 2011 Tohoku-Oki earthquake: insight from differential bathymetry" describes

measurements of seafloor deformation by the 2011 Tohoku-oki earthquake using the differential bathymetry before and after the earthquake. They compared bathymetry data collected before and after the earthquake and obtain interesting information for crustal movement. Bathymetry data captured by multi-narrow beam echo sounders equipped to research vessels were divided in narrow regions to increase resolution, and detailed difference of seafloor topography was obtained. There is a possibility that the inelastic uplift in the frontal prism deformation is related to generation of large tsunami off Sanriku region. In the Supplementary Materials, data and a method for calculation of seafloor deformation in the study are described in detail.

The requirements for the previous version have been fulfilled. They added explanation of deformation by the 2011 Tohoku-oki earthquake in the Result Section. Criterion for dividing the regions for estimation of deformation by the event and difference of slip amounts between the profiles were reasonably explained. Description about the relationship between seafloor deformation and tsunami genesis was additionally described in the Discussion Section. The data processing about removal of data with large incident angles and mathematical description were also well explained. Other comments for the previous manuscript have been reflected.

The revised manuscript has much improvement and is thought to be understandable for readers. The useful and unique method is newly proposed to estimate seafloor deformation by a large interplate earthquake. In addition, the authors give new evidence for consideration in deformation near a trench during large events. Their data processing is well described in Supplementary Materials. I feel the manuscript is acceptable for the publication.

Reply: Thanks for the constructive comments.